



# Developing an atlas of rain-induced leading edge erosion for wind turbine blades in the Dutch North Sea

Marco Caboni[1] and Gerwin van Dalum[2]

[1]TNO, Wind Energy Technology, Westerduinweg 3, 1755 LE Petten, The Netherlands
[2]Whiffle, Molengraaffsingel 8, 2629 JD Delft, The Netherlands

**Correspondence:** Marco Caboni (marco.caboni@tno.nl)

**Abstract.** To support the ongoing development of offshore wind energy in The Netherlands and to maintain the current assets, it is essential to provide wind farm operators accurate estimates of wind turbine blade erosion. Unfortunately, there is currently a shortage of information on wind turbine erosion risk, especially in offshore regions. In this work, we developed an atlas detailing rain-induced leading edge erosion for wind turbine blades in the Dutch North Sea, using weather simulations spanning a decade. These weather simulations were validated using recent offshore and onshore measurements and incorporated into a fatigue-based damage model, linking weather conditions to blades' leading edge erosion. The results reveal that the erosive impact of rainfall on wind turbine blades varies across the Dutch North Sea. The estimated average incubation period, which indicates the leading edge protection system's lifespan, ranges from 8 to 9 years in the southwestern region, decreasing to 6 to 7 years in the northeastern area. This is due to both the higher average wind speeds and greater rainfall amounts occurring in the northeastern locations compared to the southwestern ones. This paper emphasizes that the northeastern regions of the Dutch North Sea, which are being examined for potential wind farm developments post-2030, will encounter higher erosion risks compared to those currently operating in southern locations, possibly requiring enhanced mitigation strategies.

## 1 Introduction

The Dutch North Sea is undergoing significant development in wind energy, with numerous wind farms currently operating and more planned (Noordzeeloket, 2024). For developers of offshore wind farms and turbines, assessing the risk of rain-induced leading edge erosion (LEE) is crucial. Raindrops hitting the blades lead to gradual mechanical wear on the surface of their leading edge (Slot et al., 2015). Leading edge wear continues to occur despite the development and implementation of advanced leading edge protection (LEP) systems on modern wind turbine blades (Mishnaevsky et al., 2021). Damaged blades result in decreased turbine performance (Bak et al., 2020; Maniaci et al., 2020; Vimalakanthan et al., 2023; Castorrini et al., 2023), forcing wind farm operators to conduct frequent, costly, labor-intensive, and possibly hazardous maintenance procedures to fix the blades. LEE poses a significant challenge to the rapid deployment of wind energy, particularly in offshore environments.

LEE is a fatigue-driven process in which each droplet impact contributes to cumulative damage. (Slot, 2021; Mishnaevsky, 2019). The severity of LEE is tied to the frequency and speed of impacts, as well as the drop size distribution (DSD). Although





the quantity of impacts and the size of raindrops are linked to rain conditions, the speed of the impacts primarily depends on the turbine's rotational speed, which is linked to wind speed. Thus, evaluating LEE must consider the simultaneous conditions of both rain and wind speed. Caboni et al. (2024) carried out concurrent measurements of rain and wind speed at an offshore platform in the Dutch North Sea, linking the weather conditions to the accumulation of wind turbine damage. Other recent studies focused on evaluating LEE using measurements carried out at onshore and coastal areas (Hasager et al., 2020, 2021;

Shankar Verma et al., 2021; Verma et al., 2021; Letson and Pryor, 2023; Ásta Hannesdóttir et al., 2024a; Méndez et al., 2024). Numerical techniques can be utilized to bridge the gap in offshore measurements, providing essential data to assess erosion risks associated with wind energy deployment. Ásta Hannesdóttir et al. (2024b) incorporated wind turbine erosion into their analysis by creating a rain erosion atlas for the Norwegian and Danish North Sea, along with the Baltic Sea. This was achieved by linking the ERA5 and NORA3 reanalysis datasets to a wind turbine erosion model. Other studies concentrate on creating

maps of wind resources without addressing the issue of wind turbine erosion (Nawri et al., 2014; Hahmann et al., 2020; Davis et al., 2023; Larsén et al., 2022; Mortensen et al., 2014; Copernicus-Climate-Change-Service).

    The literature review reveals a significant gap in knowledge regarding the large-scale mapping of rain erosion risks for wind turbine blades, particularly in the Dutch North Sea. This study aims to address this gap by developing an atlas, based on long-term meso-scale simulations, to evaluate the risk of LEE in the Dutch North Sea. As part of this project, meso-scale

simulations were compared with both high resolution large eddy simulations (LES) and actual measurements. The benefit of using meso-scale simulations and especially LES instead of existing lower resolution reanalysis datasets is that it involves less spatial and temporal averaging of local variables, and can therefore better catch extreme precipitation events.

    The innovative numerical method used in this research relies on Whiffle's LES model ASPIRE (Atmospheric Simulation Platform for Innovation, Research, and Education). ASPIRE started as a GPU implementation of the DALES (Dutch Atmo-

spheric Large-Eddy Simulation) model (Heus et al., 2010; Schalkwijk et al., 2012) that has since received numerous improvements that allow it to be used as an operational weather model. Its main innovation is that the model's calculations are highly parallelized using the capabilities of the GPU, which cuts down the runtime by several orders of magnitude compared to traditional implementations on the CPU. Crucially, this allows for simulations on a much larger domain and/or a much higher resolution at the same computational cost. ASPIRE can also run meso-scale simulations that do not resolve any turbulence like

in LES, but benefit from the same computational speed-up.

    In the context of this work, ASPIRE is used to perform a high resolution LES of the Dutch North Sea over a period of one year, and a lower resolution meso-scale simulation over a longer period of time to capture the long-term precipitation climate. The one-year high resolution LES was employed to verify the meso-scale simulations. Both LES and meso-scale simulations were also compared with the experimental data published by Caboni et al. (2024). The weather data obtained with ASPIRE's

simulations are utilized to determine turbine erosion rates through a fatigue-based erosion model applied to a virtual 15 MW wind turbine.

    The remainder of this paper is organized into three sections. The methodology section details the models and simulations, including the methodology used to evaluate erosion damage and the measurements utilized to validate the simulations. In the





results section, we present the validation of the simulations based on one year of measurements, along with the developed atlas.
The final section will draw conclusions and outline future research directions.

## 2 Methods

### 2.1 Whiffle's ASPIRE weather simulations

ASPIRE was used to create a precipitation atlas for the Dutch North Sea by simulating weather conditions over a long period of time. A description of the model, its governing equations, and its boundary conditions can be found in Baas et al. (2023). Following the methods described by Storey and Rauffus (2024), simulations of historical weather conditions are performed by coupling large-scale data from ECMWF's ERA5 reanalysis dataset (Hersbach et al., 2020) to the lateral boundaries of a meso-scale version of the model. In a similar manner, this meso-scale simulation is in turn coupled to the boundaries of a nested high resolution LES. Both the meso-scale simulation and the LES use an implementation of the microphysics model by Grabowski (1998), which locally calculates the rate of rain droplet formation and simulates the subsequent precipitation. Since ASPIRE works with model time steps of a few seconds, it is able to capture short-lived events such as the high rain rate events that are of special interest for this study.

Within ASPIRE, moisture is treated using two prognostic variables, distinguishing between the total non-precipitating specific humidity $q_t$ consisting of water vapor and non-precipitating liquid and ice water, and precipitating water $q_r$. The former is diagnostically partitioned into its components using an all-or-nothing scheme that assumes the water to be homogeneously distributed over a grid cell. As such, grid cells with $q_t$ below the local saturation mixing ratio $q_{sat}$ contain no clouds, whereas any $q_t$ above saturation is immediately interpreted as non-precipitating liquid or ice water, depending on the temperature. Following Grabowski (1998), this cloud water content is subsequently used to calculate autoconversion and accretion rates for droplet formation, as well as deposition and evaporation rates. Finally, precipitation of the formed droplets is accounted for by an additional advective term that makes the droplets fall. The resulting change in precipitating water content $q_r$ is subtracted from $q_t$, such that the total amount of water is conserved. Note that although ASPIRE works with a single prognostic precipitation variable $q_r$, different species (most importantly rain and snow) are diagnosed using a temperature-based partitioning, each with their own version of the DSD, mass-diameter relation, and fall velocity. However, as this study is specifically about rain-induced LEE, the remainder of this section focuses solely on rain.

To determine the incubation period, information about the rain DSD is required. This information can be extracted from the simulations by utilizing the fact that the underlying microphysics model assumes the Marshall-Palmer distribution (Marshall and Palmer, 1948):

$$N(D) = N_0 \exp(-\lambda D), \tag{1}$$

where $N(D)$ is the DSD in $m^{-4}$ as a function of drop diameter $D$, $\lambda$ is a shape parameter, and $N_0$ is a fixed prefactor. The shape parameter $\lambda$ depends on the local amount of rain water $q_r$ (in kg kg$^{-1}$), the local air density $\rho$, and the density of water



$\rho_w$ through the following relation:

$$\rho q_r = \int\limits_0^\infty \mathrm{d}D \frac{\pi}{6} D^3 \rho_w \cdot N(D) = \pi \rho_w N_0 \lambda^{-4}. \tag{2}$$

Combined with the fall velocity used in ASPIRE (Lin et al., 1983),

$$v_t(D) = aD^b \left( \frac{\rho_0}{\rho} \right)^{1/2}, \tag{3}$$

where $a$, $b$ and $\rho_0$ are empirical constants, the rain rate RR can be calculated as a function of the shape parameter $\lambda$ by using its definition:

$$\mathrm{RR} = \int\limits_0^\infty \mathrm{d}D \frac{\pi}{6} D^3 \cdot N(D) \cdot v_t(D) = \frac{\pi}{6} N_0 a \left( \frac{\rho_0}{\rho} \right)^{1/2} \lambda^{-(4+b)} \Gamma(4+b), \tag{4}$$

with $\Gamma(x)$ being the gamma function. As a best estimate for the DSD from the simulation, we assume the rain rate to be constant over the output interval, and use its value to determine the shape parameter (and by extension the corresponding DSD) by inverting Equation 4. The reason we do this is because rain-related quantities such as $\lambda$ and the rain rate can vary strongly on short time scales, while simulation output is always an average over a certain period of time and not an instantaneous value. This method of dealing with the consequences of temporal averaging guarantees that the estimated DSD matches the total precipitation during a given time interval, which is not the case with most other methods due to the nonlinearity of the equations involved. Since the rain rate fluctuates constantly, the accuracy of this estimated DSD increases with decreasing output intervals.

To create a long-term atlas of the precipitation climate over the Dutch North Sea, two simulations have been performed. The first is a high resolution 1-year LES (March 2022 - March 2023) covering most of the Dutch North Sea, serving as our baseline. In addition, a larger but much lower resolution stand-alone meso-scale simulation was done, covering a period of 10 years (2014 – 2023) with a domain size of over 1,000 kilometers. The goal of the latter is to provide data for an atlas that spans a long period of time, while the former is a shorter but much higher fidelity simulation that can be used to verify the latter and correct for its shortcomings. The settings used for the simulations are summarized in Table 1.

All simulations were performed "in series" on a per-month basis, i.e. the next simulation starting from the end state of the previous simulation, so only the first day of a month required spin-up time. Furthermore, the output, consisting of two-dimensional grids of time series for multiple variables, most importantly includes rain rate and wind speed. The output is always sampled at 100 m elevation, and it is sampled locally (no spatial averaging over multiple grid points). Although the output resolution is almost identical between the LES and the meso-scale simulation, output data points therefore represent an area of 120 m x 120 m in the former, and an area of 2 km x 2 km in the latter. As such, combined with the different output frequencies (see Table 1), the output of the meso-scale simulation is more smoothed than the LES in a spatial sense as well as a temporal one.





**Table 1.** Summary of the settings used in the simulations. Note that the vertical levels are non-uniformly distributed.

|  | High resolution 1-year LES | Stand-alone 10-year meso-scale simulation |
| --- | --- | --- |
| Period | 21/03/2022 - 20/03/2023 | 01/01/2014 - 31/12/2023 |
| Dimensions | 245.76 km x 384 km x 8 km | 1,024 km x 1,024 km x 8 km |
| Horizontal resolution | 120 m x 120 m | 2 km x 2 km |
| Number of vertical levels | 64 | 128 |
| Output sampling method | 1-minute averages | hourly averages |
| Output resolution | 1920 m x 1920 m | 2 km x 2 km |

## 2.2 Accumulated damage estimation

120  The erosion process is influenced by the fatigue properties of the blade's LEP systems, as well as the size and number of droplets impacting the surface at a specific speed. Wear particle emission from the leading edge begins once the incubation period ends. Typically, the incubation period is considered the LEP system's lifespan (Slot, 2021). In this study, we estimated the incubation period using the "ASTM - Multiple linear regression fit equations," which include a modified dependence on drop size (Slot et al., 2025). This model applies only to liquid drop impacts. Heymann (1979) developed multiple linear

125  regression fit equations for the incubation life based on an extensive ASTM test program, where each material is characterized by its normalized incubation resistance number (NOR). To determine the NOR for current LEP systems on wind turbine blades, we conducted a literature review on rotating arm rain erosion tests on LEP systems (Caboni et al., 2025). The NOR values for current LEP systems range from 0.001 to 0.033. For details on the equations used, the reader is referred to Caboni et al. (2024). To consider the cumulative impact of varying rain and wind speed conditions on the erosion process, assuming linear damage

130  accumulation, the Palmgren-Miner's rule was applied. The accumulated damage is represented by a parameter $F$, which starts at zero at the onset of the erosion process and reaches one at the conclusion of the incubation period.

For simplicity, this study concentrated on the accumulated damage at the blade tips of the virtual IEA 15 MW reference wind turbine (Gaertner et al., 2020), which have a maximum tip speed of 95 m/s and utilize a polyurethane LEP system with an estimated NOR of 0.003. The cumulative damage is closely linked to the relative impact velocity between the blade and

135  the raindrops. This velocity is influenced by the blade's rotational speed, the fall velocity of the droplets, and the aerodynamic interactions between the rain and the wind (Barfknecht and von Terzi, 2023). For simplicity in this work, we assumed that the fall velocity of the raindrops is negligible compared to the blade tip speed and that there are no aerodynamic interactions between the droplets and the wind. Therefore, we considered the impact speed between the blade and the raindrops to be equal to the tangential tip speed. The tangential tip speed is determined by the rotor speed, which is connected to the wind speed.

140  The current maximum tip speed of modern offshore wind turbines is around 90 m s$^{-1}$ (Wind Energy - The Facts, 2024).



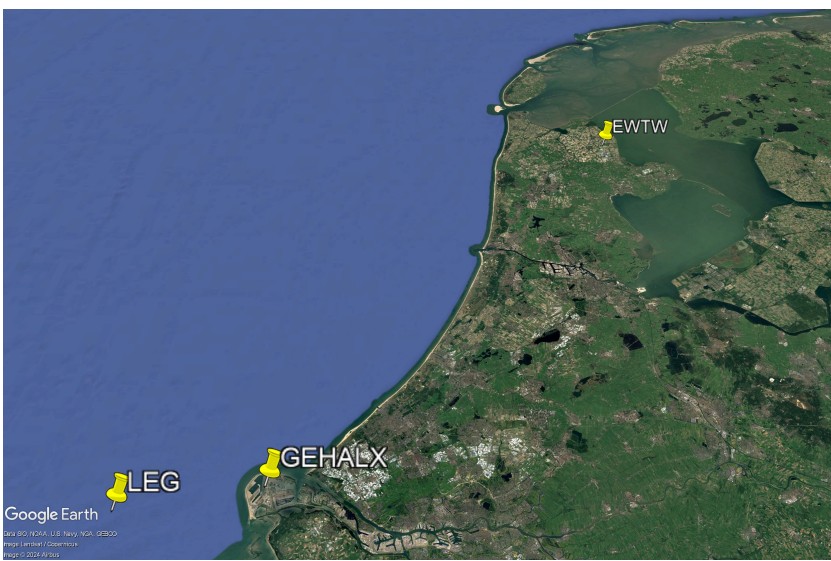

**Figure 1.** Map illustrating the locations in The Netherlands where simultaneous wind and rain measurements were conducted as part of the PROWESS project (Caboni et al., 2024). Image courtesy of © Google Earth.

## 2.3 Rain and wind speed measurements

From March 2022 to March 2023, TNO conducted simultaneous measurements of rainfall and wind speed at three locations in The Netherlands (Caboni et al., 2024). These sites included an offshore location (LEG), a coastal location (GEHALX), and an onshore location (EWTW), as shown in Figure 1. Rainfall was measured using OTT Parsivel$^2$ disdrometers at LEG and GEHALX, and a Thies LPM disdrometer at EWTW. Wind speed was recorded with cup anemometers at EWTW and GEHALX, and a Leosphere WindCube V2 LiDAR at LEG. To focus on rain-induced erosion, events involving snow and hail were excluded, leaving only rainy periods. Rainfall data was collected at a frequency of 1 Hz, while wind speed data was recorded at 0.1 Hz.

For each observation interval, we calculated the rain parameters like intensity and amount by using the measured DSD and droplet fall velocity using the formulas presented by Tilg et al. (2020). However, it is essential to note that DSD measurements obtained from current sensors, such as disdrometers, remain quite uncertain (Letson and Pryor, 2023; Caboni et al., 2024; Ásta Hannesdóttir et al., 2024a). This is due to the fact that these sensors and their algorithms are typically optimized to accurately detect total precipitation amounts rather than the DSD itself. Additionally, uncertainties in sensor calibration and potential measurement errors caused by wind turbulence, insects, or, at offshore locations, sea spray contribute to the overall uncertainty in disdrometer measurements.



# 3 Results

In this section we show the comparison between measurements and simulations and present the LEE atlas. The comparison was conducted over the course of a year using high resolution LES and stand-alone meso-scale simulation. The atlas was developed by using 10-year meso-scale simulations. We utilized meso-scale simulations to develop the atlas because they are more computationally affordable than LES and, as noted below, they provide a fairly good estimate of trends for parameters related to leading edge erosion across the studied locations.

## 3.1 Comparison between measurements and ASPIRE

The comparison between measurements and simulations is presented in two sections. The first section deals with the comparison of aggregated figures related to erosion, such as rain amount and accumulated damage, obtained using measurements, LES, and meso-scale simulations. The second section focuses on a more detailed comparison between measurements and simulations, including DSD and droplet fall velocities. For brevity, we only compared measurements with LES in this section.

### 3.1.1 Comparison of aggregate erosion figures between measurements, high resolution LES, and stand-alone meso-scale simulation

In this section, numerical results and measurements over one year are compared at the three aforementioned locations, estimating aggregate erosion figures such as yearly rainfall and accumulated damage. Numerical data and measurements are utilized to assess the accumulated tip damage of the IEA 15 MW reference wind turbine, virtually installed at the specified locations. It is assumed that the turbine blades utilize a polyurethane LEP system. Table 2 presents a comparison of aggregated meteorological and erosion data between simulations and experiments at the three locations.

LES overestimate the average wind speed during rainy events at LEG and EWTW by up to 6%, while at GEHALX, it is underestimated by 8%. The predicted rain amount is up to 13% lower than the measurements at LEG and EWTW, while at GEHALX, it is 13% higher. To provide further insight into the wind speed comparison, Figure 2 displays the measured and LES simulated probability distribution functions for wind speed during rainy events. By utilizing the measured annual wind speed and rainfall, we calculated the yearly accumulated damage. Taking the reciprocal of this accumulated damage provides an estimate of the incubation period (in years). As seen in Table 2, LES underestimate the accumulated damage by 23% at EWTW, 45% at GEHALX, and 66% at LEG.

Compared to measurements, the meso-scale average wind speed during rain is underestimated by 2% at LEG and 10% at GEHALX, while it is overestimated by about 4% at EWTW. The meso-scale rain amount is underestimated by 4% at LEG and 8% at EWTW, but overestimated by approximately 21% at GEHALX. Consequently, meso-scale simulations underestimate accumulated damage by 29% at EWTW, 55% at GEHALX, and 70% at LEG.

With respect to accumulated damage, the results from LES align more closely with experimental data than those from meso-scale simulations. Our analysis, which is not included here for brevity, suggests that, although not perfectly aligned with measurements, LES is more effective at capturing extreme events, namely those characterized by high intensity and large wind



speeds. The relative variations in accumulated damage estimated using either LES or meso-scale simulations and experiments highlight the significant uncertainty in accurately determining the absolute value of this parameter. We leave the task of reducing these uncertainties to future research. However, as demonstrated in Table 2, both LES and meso-scale simulations effectively capture trends in erosion-related parameters across the locations. Therefore, we consider meso-scale simulations a suitable method for capturing erosion risks across the Dutch North Sea, and thus, they are used to develop the long-term atlas.

**Table 2.** Comparison between measured and simulated average wind speed during rainy intervals, total rainfall, accumulated damage, and estimated incubation period, based on one year of measurements from March 21, 2022, to March 21, 2023. The figure shows ASPIRE's numerical results from the high resolution LES and stand-alone meso-scale simulation.

| location | source | WS [m s$^1$] | RA [mm] | F [-] | IP (= F$^{-1}$) [yr] |
|---|---|---|---|---|---|
| LEG | meas. | 11.21 | 555 | 0.43 | 2.34 |
|  | ASPIRE (LES) | 11.60 (3.49 %) | 499 (-10.07 %) | 0.15 (-65.77 %) | 6.84 (192.11 %) |
|  | ASPIRE (meso-scale) | 10.99 (-1.93 %) | 530 (-4.42 %) | 0.12 (-70.75 %) | 8.00 (241.90 %) |
| EWTW | meas. | 8.56 | 591 | 0.18 | 5.54 |
|  | ASPIRE (LES) | 9.08 (6.01 %) | 512 (-13.44 %) | 0.14 (-23.33 %) | 7.23 (30.43 %) |
|  | ASPIRE (meso-scale) | 8.87 (3.63 %) | 542 (-8.42 %) | 0.13 (-29.12 %) | 7.82 (41.09 %) |
| GEHALX | meas. | 10.82 | 457 | 0.25 | 3.93 |
|  | ASPIRE (LES) | 9.96 (-7.98 %) | 520 (13.83 %) | 0.14 (-44.56 %) | 7.09 (80.38 %) |
|  | ASPIRE (meso-scale) | 9.72 (-10.14 %) | 555 (21.44 %) | 0.11 (-55.20 %) | 8.77 (123.20 %) |

### 3.1.2 Detailed comparison between measurements and high resolution LES

To explore the aforementioned differences between simulations and measurments in total accumulated damage in greater detail, we present the annual rainfall, total damage and accumulated damage per rain amount organized into bins based on wind speed and rain rate in Figures 3, 4 and 5, respectively. The findings in this section remain consistent whether we consider LES or meso-scale simulations. The primary difference, as previously explained, is that LES captures more extreme events compared to meso-scale simulations. Therefore, for the sake of brevity, we will conduct this analysis using only the LES.

Damage accumulation is more significant during extreme events, namely high rain intensity and strong wind speeds. Caboni et al. (2024) noted that at LEG 30% of the yearly damage is accumulated over just 12 hours, in which the wind speed is greater than 17.5 m s$^{-1}$ and the rain rate is greater than 7.5 mm h$^{-1}$. In fact, high-intensity events feature a greater number of relatively large droplets compared to low-intensity events. Larger droplets are more erosive than smaller ones when comparing the same volume of liquid (Barfknecht and von Terzi, 2024; Slot et al., 2025). As noted earlier, wind speed also influences LEE. At higher wind speeds, the blades rotate faster, resulting in a greater impact speed between the rain droplets and the blades. Figures 3 and 4 show that the extreme events differ between measurements and simulations, offering a first explanation for the



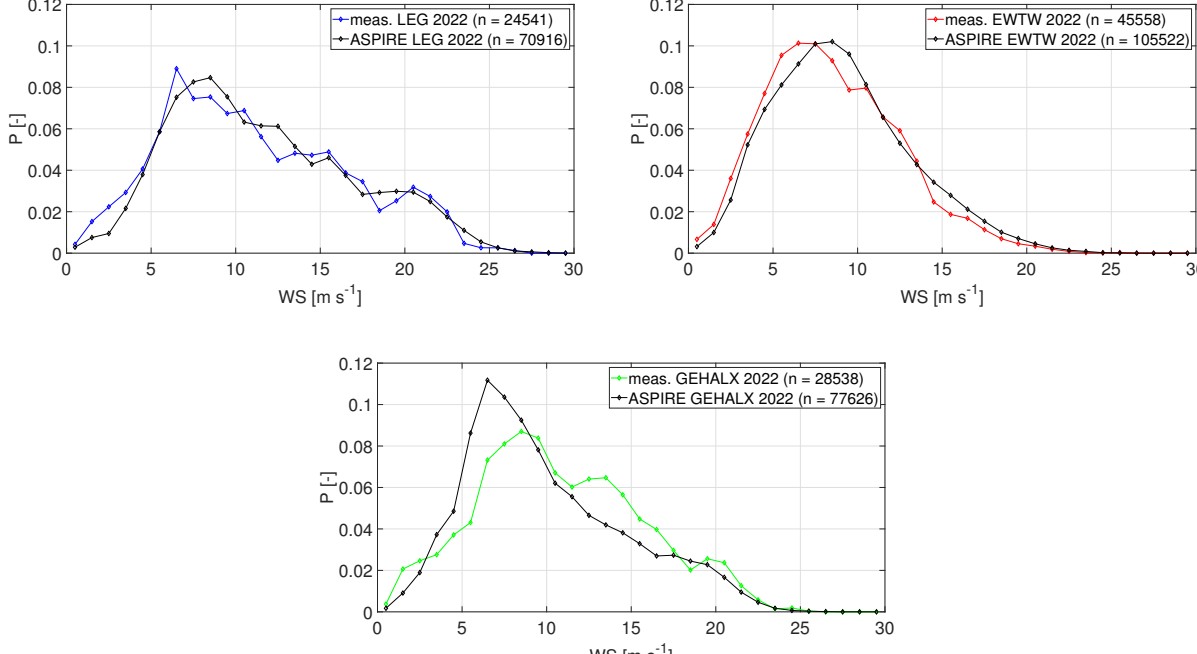

**Figure 2.** Comparison of measured and simulated probabilities of wind speed during rainy intervals at the measurement sites. Here, the numerical results are from the high resolution LES.

variations in accumulated damage. Except for GEHALX, both LEG and EWTW have recorded more extreme events than those predicted by simulations. The difference in extreme events is especially pronounced at LEG, resulting in significant variation in accumulated damage. Currently, the authors cannot explain why extreme events are more underestimated at LEG compared to other locations. It is possible that this discrepancy is related to the aforementioned uncertainties affecting disdrometer

measurements. Compared to onshore locations, offshore and coastal environments are experimentally shown to experience more extreme rain events (Caboni et al., 2024; Hasager et al., 2020). This trend also appears to be captured by the simulations (see Figure 3).

In addition to extreme events, variations in DSD also contribute to differences in damage accumulation. The impact of these DSD differences is illustrated in Figure 5, which shows accumulated damage per unit of rainfall. The tables for each bin display

the accumulated damage corresponding to every millimeter of rain that has fallen. It is noted that, within the same categories, the observed accumulated damage per millimeter of rain exceeds the predictions made by the simulations. This discrepancy is linked to the DSD.



**RA [mm], meas. LEG 2022 (n = 24541)**

| WS [m s⁻¹] | 0.5 | 3 | 7.5 | 15 | >20 | Σ |
|---|---|---|---|---|---|---|
| 2.5 | 9.928 | 18.35 | 6.987 | 7.867 | 17.93 | 61.06 |
| 7.5 | 31.89 | 54.49 | 18.37 | 13.05 | 7.036 | 124.8 |
| 12.5 | 21.29 | 55.67 | 26.7 | 23.69 | 23.43 | 150.8 |
| 17.5 | 14.5 | 44.39 | 32.66 | 35.16 | 19.03 | 145.7 |
| 22.5 | 7.387 | 23.58 | 16 | 20.39 | 2.662 | 70.02 |
| 27.5 | 0.3076 | 1.763 | 0.2039 | 0 | 0 | 2.275 |
| >30 | 0 | 0 | 0 | 0 | 0 | 0 |
| Σ | 85.3 | 198.2 | 100.9 | 100.2 | 70.08 | 554.7 |

RR [mm h⁻¹]

**RA [mm], ASPIRE LEG 2022 (n = 70916)**

| WS [m s⁻¹] | 0.5 | 3 | 7.5 | 15 | >20 | Σ |
|---|---|---|---|---|---|---|
| 2.5 | 4.299 | 14.94 | 8.75 | 4.798 | 0.3496 | 33.14 |
| 7.5 | 27.58 | 72.55 | 20.48 | 11.92 | 0.7256 | 133.3 |
| 12.5 | 39.62 | 60.82 | 26.5 | 12.12 | 3.545 | 142.6 |
| 17.5 | 32.94 | 54.52 | 21.8 | 9.905 | 1.264 | 120.4 |
| 22.5 | 15.85 | 41.23 | 8.602 | 0.7403 | 0 | 66.43 |
| 27.5 | 1.191 | 1.637 | 0.1586 | 0 | 0 | 2.987 |
| >30 | 0.004303 | 0 | 0 | 0 | 0 | 0.004303 |
| Σ | 121.5 | 245.7 | 86.29 | 39.49 | 5.884 | 498.8 |

RR [mm h⁻¹]

**RA [mm], meas. EWTW 2022 (n = 45558)**

| WS [m s⁻¹] | 0.5 | 3 | 7.5 | 15 | >20 | Σ |
|---|---|---|---|---|---|---|
| 2.5 | 21.03 | 58.28 | 22.62 | 11.02 | 3.735 | 116.7 |
| 7.5 | 60.94 | 137.8 | 46.12 | 18.9 | 5.434 | 269.2 |
| 12.5 | 40.03 | 90.75 | 23.83 | 13.25 | 3.775 | 171.6 |
| 17.5 | 8.222 | 18.14 | 2.101 | 0.8686 | 0 | 29.33 |
| 22.5 | 0.875 | 2.62 | 0.7243 | 0.2083 | 0 | 4.428 |
| 27.5 | 0.03058 | 0.02309 | 0 | 0 | 0 | 0.05367 |
| >30 | 0.02065 | 0.02814 | 0 | 0 | 0 | 0.04879 |
| Σ | 131.1 | 307.7 | 95.39 | 44.24 | 12.94 | 591.4 |

RR [mm h⁻¹]

**RA [mm], ASPIRE EWTW 2022 (n = 105522)**

| WS [m s⁻¹] | 0.5 | 3 | 7.5 | 15 | >20 | Σ |
|---|---|---|---|---|---|---|
| 2.5 | 10.58 | 20.45 | 2.968 | 1.096 | 1.167 | 36.25 |
| 7.5 | 44.98 | 84.79 | 21.22 | 4.75 | 2.425 | 158.2 |
| 12.5 | 59.73 | 115.7 | 34.71 | 13.76 | 0.3519 | 224.2 |
| 17.5 | 22.59 | 47.63 | 9.686 | 3.707 | 0.3377 | 83.95 |
| 22.5 | 3.061 | 5.978 | 0.1849 | 0 | 0 | 9.224 |
| 27.5 | 0.04337 | 0.04576 | 0 | 0 | 0 | 0.08913 |
| >30 | 0 | 0 | 0 | 0 | 0 | 0 |
| Σ | 141 | 274.6 | 68.78 | 23.31 | 4.282 | 511.9 |

RR [mm h⁻¹]

**RA [mm], meas. GEHALX 2022 (n = 28538)**

| WS [m s⁻¹] | 0.5 | 3 | 7.5 | 15 | >20 | Σ |
|---|---|---|---|---|---|---|
| 2.5 | 9.835 | 34.16 | 14.87 | 7.214 | 8.431 | 74.51 |
| 7.5 | 40.26 | 67.82 | 20.67 | 14.68 | 15.73 | 159.2 |
| 12.5 | 32.76 | 62.82 | 18.11 | 14.16 | 7.843 | 135.7 |
| 17.5 | 17.76 | 39.94 | 6.634 | 4.335 | 2.721 | 71.39 |
| 22.5 | 6.371 | 8.777 | 0.7887 | 0 | 0 | 15.94 |
| 27.5 | 0.074 | 0.06001 | 0 | 0 | 0 | 0.134 |
| >30 | 0 | 0 | 0 | 0 | 0 | 0 |
| Σ | 107.1 | 213.6 | 61.07 | 40.39 | 34.72 | 456.8 |

RR [mm h⁻¹]

**RA [mm], ASPIRE GEHALX 2022 (n = 77626)**

| WS [m s⁻¹] | 0.5 | 3 | 7.5 | 15 | >20 | Σ |
|---|---|---|---|---|---|---|
| 2.5 | 8.08 | 14.03 | 5.031 | 7.744 | 0.3482 | 35.23 |
| 7.5 | 41.92 | 87.25 | 28.02 | 17.83 | 3.075 | 178.1 |
| 12.5 | 40.55 | 65.98 | 26.77 | 19.55 | 4.285 | 157.1 |
| 17.5 | 25.67 | 55.95 | 17.46 | 12.93 | 5.554 | 117.6 |
| 22.5 | 8.257 | 17.25 | 1.857 | 1.871 | 2.205 | 31.43 |
| 27.5 | 0.1416 | 0.3932 | 0 | 0 | 0 | 0.5348 |
| >30 | 0 | 0 | 0 | 0 | 0 | 0 |
| Σ | 124.6 | 240.8 | 79.13 | 59.94 | 15.47 | 520 |

RR [mm h⁻¹]

**Figure 3.** Annual rainfall categorized into bins based on wind speed and rain rate. Presented here are the numerical results from the high resolution LES.





**Figure 4.** Total annual damage classified into categories based on wind speed and rain rate. Presented here are the numerical results from the high resolution LES.





**Figure 5.** Yearly accumulated damage per rain amount sorted out into wind speed and rain rate bins. Presented here are the numerical results from the high resolution LES.





Figure 6 shows the comparison between the measured and simulated DSD at the three locations, for a fixed rain rate and varying wind speeds. As mentioned above, the simulations assume a DSD based on a Marshall-Palmer distribution. The com-

parison shows that the Marshall-Palmer distribution matches the measured DSD for drop sizes up to approximately 3 mm. Beyond this size, the Marshall-Palmer distribution significantly underestimates the droplet amount. This difference becomes more pronounced as the wind speed increases. Indeed, the measurements clearly indicate that wind speed influences the DSD. At higher wind speeds, a greater number of relatively large droplets are observed. Similar results were also observed by Montero-Martínez and García-García (2016) and Thurai et al. (2019).

**Figure 6.** Mean drop size distributions for a fixed rain rate of 3 mm h$^{-1}$ and different wind speeds. ASPIRE's results are here derived from the LES data.





To finalize the comparison between observations and simulations, we also include the drop fall velocity. Figure 7 presents the comparison between the measured and simulated drop fall velocities at the three locations, for a fixed rain rate and different wind speeds. Concerning fall velocity, the approximation used in ASPIRE is based on a simple model provided by Lin et al. (1983). The approximation by Lin et al. (1983) aligns well with measurements for droplets with diameters of up to 2 mm. Beyond this value, the measurements show a constant falling speed, whereas Lin et al. (1983)'s approximation assumes that

the fall velocity increases. Similar to the DSD, the measured fall speed also appears to vary with wind speed, showing lower fall speeds at higher wind velocities.

**Figure 7.** Median drop fall velocity for a fixed rain rate of 3 mm h$^{-1}$ and different wind speeds. ASPIRE's results are here derived from the LES data.





## 3.2 Dutch North Sea wind turbine blades' rain-induced leading edge erosion atlas

To reduce the demand for computational resources, meso-scale simulations are conducted to develop the long-term atlas. As discussed in Section 3.1.1, we conducted a thorough comparison between LES and meso-scale simulations over a year, which
revealed that LES is more effective at capturing extreme events, achieving a closer alignment with experimental results. As shown in Table 2, in terms of wind speed and rain amount, the LES and meso-scale models exhibited differences of up to 5%, while the discrepancy in accumulated damage reached 25%. Although meso-scale simulations are therefore not as accurate as LES, we had concluded that they can still be considered adequate for developing the long-term atlas as they can capture trends in erosion parameters correctly.

Figures 8, 9 10 and 11 illustrate contour maps of the annual mean wind speed at 100 meters above the ground and mean sea level, annual rainfall, annual accumulated damage, and annual incubation period, respectively. The contour maps presented here are derived by averaging the yearly simulation results over a 10-year period, from 2014 to 2023. Simplified coastlines are depicted by solid black lines on the maps, while dashed lines indicate the boundaries of the Dutch North Sea. Current, planned, and (estimated) future wind farm areas (North-Sea-Energy, 2024) are represented by blue, red, and green lines, respectively.

Figure 9 shows that the incubation period within the Dutch North Sea varies. It ranges from 8 to 9 years in the southwest and decreases to 6 to 7 years as one moves towards the northeast.  The variation in the incubation period is linked to differences in mean wind speed and annual rainfall. As shown in Figure 10, the wind speed is approximately 9.6 m/s in the south and increases to about 10.6 m/s in the northeast. Additionally, the northeast experiences higher rainfall, up to 600 mm annually, compared to 500 mm in the south, as shown in Figure 11.

Therefore, in the northeastern part of the Dutch North Sea, which is a potential area for future wind farms, developers will need to consider the greater impact of LEE compared to wind farms in the southern regions. The reason for this is that these areas experience higher wind speeds and greater rainfall. To clarify the differences between the southern and northern areas, Figure 12 presents a comparison of the simulated annual rainfall and accumulated damage at two locations: one at the LEG offshore platform (situated in the southwest) and the other in the northeast, located at coordinates 53.9968, 6.0106. It appears
that in the southwest there was more rainfall with relatively higher intensity, whereas in the northeast, more rain fell but with relatively lower intensity. However, in the northeast, more rain falls at higher wind speeds, which contributes to greater erosivity in that region.

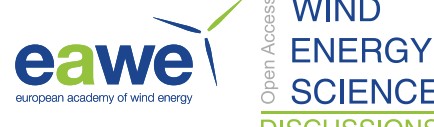

**Figure 8.** Contour map of yearly accumulated damage based on 10 years of meso-scale simulations. Zones of currently operational wind farms are depicted in blue, while areas of wind farms planned to be operational before 2030 and search areas for wind farms to be commissioned after 2030 are depicted in red and green, respectively. Dashed lines depict the boundaries of the Dutch North Sea.



**Figure 9.** Contour map of yearly incubation period based on 10 years of meso-scale simulations. Zones of currently operational wind farms are depicted in blue, while areas of wind farms planned to be operational before 2030 and search areas for wind farms to be commissioned after 2030 are depicted in red and green, respectively. Dashed lines depict the boundaries of the Dutch North Sea. The color scale of this map is adapted to highlight differences across the Dutch North Sea. This resulted in nearly constant coloring in other areas, which should not be interpreted as constant values, but rather as values outside the selected scale.



**Figure 10.** Contour map of yearly mean wind speed at 100 m above the ground and mean sea level based on 10 years of meso-scale simulations. Zones of currently operational wind farms are depicted in blue, while areas of wind farms planned to be operational before 2030 and search areas for wind farms to be commissioned after 2030 are depicted in red and green, respectively. Dashed lines depict the boundaries of the Dutch North Sea.



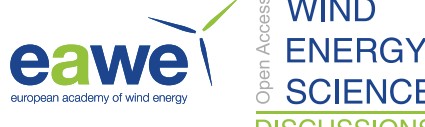

**Figure 11.** Contour map of yearly rain amount based on 10 years of meso-scale simulations. Zones of currently operational wind farms are depicted in blue, while areas of wind farms planned to be operational before 2030 and search areas for wind farms to be commissioned after 2030 are depicted in red and green, respectively. Dashed lines depict the boundaries of the Dutch North Sea.





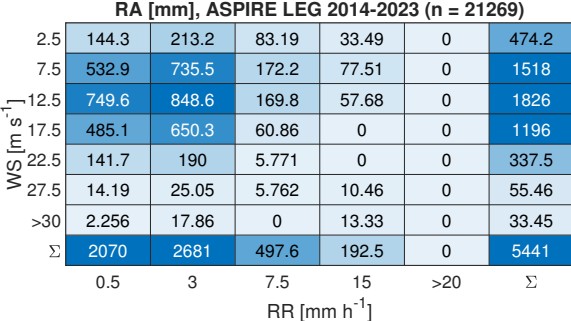

**Figure 12.** Yearly rain amount and accumulated damage sorted out into wind speed and rain rate bins. This figure compares a location situated in the southern part of the Dutch North Sea (LEG platform) with a location in the northeast (with coordinates 53.9968, 6.0106). The ASPIRE results presented in this figure are obtained from the meso-scale simulations.

# 4    Conclusions

Weather simulations carried out over a decade showed that the average erosivity of rainfall on wind turbine blades increases
from the southwestern part of the Dutch North Sea to the northeastern region. Indeed, the simulations indicate that the north-
eastern regions are characterized by both higher average wind speeds and a larger amount of rainfall. These results suggest that
future wind farms developed in the northeast are likely to encounter higher erosion rates compared to those currently operating
in the southwest. This requires special attention when developing mitigation strategies, such as using advanced leading-edge
protection systems and implementing erosion-safe modes. This mode involves reducing rotor speed during extreme events to
prolong the blades' lifespan.

A comparison of simulations and measurements at selected sites in The Netherlands showed that the accumulated damage
estimated from simulations is lower than that obtained from actual weather data. These differences can be attributed to two main
factors. Firstly, there are more extreme events recorded than those predicted by simulations. Secondly, the Marshall-Palmer
distribution, assumed by the simulations, tends to underestimate the quantity of larger droplets. Together, these factors result in
simulations predicting less accumulated damage. The relative variations in accumulated damage estimated using simulations
and experiments highlight the significant uncertainty in accurately determining the absolute value of this parameter. However,
simulations are shown to effectively capture trends in erosion-related parameters across the measured locations. Therefore, we
consider these simulations a suitable method for capturing erosion risks across the Dutch North Sea.

Future work will involve implementing a more representative drop size distribution and fall velocity in ASPIRE. In this
context, models will need to be validated with more reliable measurements, which is also a topic of ongoing and future
research. To mitigate erosion on wind turbines, high resolution weather models could be employed to investigate a now-cast
based erosion-safe mode, where the model could forecast extreme events, enabling operators to adjust rotor speed accordingly.

*Model availability.*    This research used the Whiffle proprietary LES model ASPIRE, that is commercially available as Software as a Service
(SaaS) solution from Whiffle.

*Data availability.*    The simulation data underlying this paper can be provided upon request.

*Author contributions.*    **M. C.**: Contributed in the conception and design of the study, developed the methodology for evaluating leading
edge erosion using weather data, and conducted the analysis of the results. **G. van D.**: Implemented precipitation in the weather model and
performed the simulations.

*Competing interests.*    The authors declare that there are no conflicting interests.



*Acknowledgements.* Linked to the PRecipitation atlas for Offshore Wind blade Erosion Support System (PROWESS) project, this research was supported by The Netherlands Enterprise Agency (RVO), part of the Dutch Ministry of Economic Affairs, under grant number HER+-00900701. EWTW is acknowledged for providing measurements from meteorological mast 6 at their test site in Wieringermeer. GE is acknowledged for providing measurements from meteorological mast MMX at the SIF site in Maasvlakte Rotterdam.



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
