# Peer review of "Developing an atlas of rain-induced leading edge erosion for wind turbine blades in the Dutch North Sea"

_Wind Energy Science, 2024_

## Author Comment (AC1)

Response to Referee 1

Developing an atlas of rain-induced leading edge erosion for wind turbine blades in the Dutch North Sea (wes-2024-174)

Dear Reviewer,

Thank you for reviewing our article. We have carefully addressed your comments, and the details are provided below. We sincerely appreciate your valuable comments and suggestions, which have played an important role in improving the manuscript. Your insights have helped us identify numerous shortcomings, which we hope we have satisfactorily addressed. Our revisions to the paper include significant changes, such as:

- Enhanced clarity on the novel aspects of our paper, placing them within the context of existing literature.
- A new section focusing on a detailed comparison between meso-scale simulations and high-resolution LES.
- Reformulation of the results.
- Added a Discussion section.

Sincerely,

Marco Caboni and Gerwin van Dalum

Specific comments:

In the introduction, the authors mention that "The literature review reveals a significant gap in knowledge regarding the large-scale mapping of rain erosion risks for wind turbine blades, particularly in the Dutch North Sea." A large part of the Dutch North Sea is however covered in the erosion atlas based on the NORA3 simulations from Hannisdottir et al 2024b, which is cited in the previous paragraph. The authors also critique the Hannisdottir study based on the course resolution of reanalysis data, but in the end they do not utilize the high-resolution data that they themselves generate. There seems to be no need for this study, because the NORA3 reanalysis covers the Dutch North Sea, and such data could easily have underpinned the claimed main conclusion of the study.

We have outlined the novel aspects of our paper and placed them within the context of existing literature in the introduction section as follows:

*The literature review reveals a significant gap in knowledge regarding the large-scale mapping of rain erosion risks for wind turbine blades, particularly in the Dutch North Sea. Hannesdóttir et al. (2024b) incorporated wind turbine erosion into their analysis by creating a rain erosion atlas for the Norwegian and Danish North Sea, along with the Baltic Sea. This was achieved by linking the ERA5 and NORA3 reanalysis datasets to a wind turbine erosion model. Although focusing on the Norwegian and Danish North Sea, the erosion atlas by Hannesdóttir et al. (2024b) covers part of the Dutch North Sea as well, but it is insufficient to determine the differences within the region. This is because it does not encompass a significant portion of the Dutch North Sea, particularly the central and southern areas where most current Dutch wind farms are situated. This study aims to fill this gap by creating an atlas based on long-term meso-scale simulations to assess the risk of LEE in the Dutch North Sea. By spanning 10 years to capture the long-term precipitation climate, this study surpasses the erosion atlas by Hannesdóttir et al. (2024b),*

*which was developed over 5 years. Additionally, it offers slightly better spatial resolution (2 km instead of 3 km) while maintaining the same temporal resolution (1 hour).*

*As part of this project, we aimed to identify potential improvements for simulations used to determine erosion atlases. To achieve this goal, we compared our meso-scale simulations with a high-resolution large eddy simulation (LES) setup over the course of a year over the Dutch North Sea. The advantage of using LES over meso-scale simulations is that it involves less spatial and temporal averaging of local variables, allowing it to better capture extreme precipitation events. LES simulations indeed were carried out with a spatial resolution of 120 meters and a temporal resolution of one minute. The drawback of LEE is that it requires significantly more computational resources than mesoscale simulations. To date, no such simulation and comparison has been published. To gain a deeper understanding of the simulations, LES were also compared with the experimental data published by Caboni et al. (2024), introducing an additional novel aspect.*

Further, the authors should very clearly state the literature regarding all rain erosion atlases to help the reader understand their contribution.

We have expanded the relevant literature and referenced the recent IEA Wind Task 46 Report titled "A roadmap for producing wind turbine blade coating leading edge erosion atlases: Preliminary results." This report provides a complete overview of the various meteorological data sets that have been employed in the development of erosion atlases.

Although not explicitly mentioned as an objective, I believe that the authors wish to formulate an alternative way of constructing a rain erosion atlas than what has been done before. By not acknowledging this objective, and by not comparing their detailed methods with already published methods, a reader cannot understand what the new methodology brings to the field. The authors should focus on the complementary aspects of a new methodology and what the difference between their results and already published data can teach the community regarding direction forwards for the development of erosion risk atlases. (They could also focus it on the pros and cons of the LES based approach in comparison with re-analysis or meso-scale simulations, which is currently not covered at sufficient depth in the study.)

We have included a new section that focuses on a detailed comparison between meso-scale simulations and high-resolution LES performed over 1 year over the Dutch North Sea. In this section, we specifically compare annual rainfall and total annual damage, categorized into bins based on wind speed and rain rate, and analyze the contour maps. The year-long comparison of meso-scale simulations and high-resolution large eddy simulations (LES) at selected sites in The Netherlands showed that the accumulated damage estimated from meso-scale simulations is 7 to 20% lower than that obtained from LES. These differences can be attributed to the LES setup's ability to capture more extreme events due to its finer spatial and temporal resolution. Moreover, the side-by-side comparison of the contour maps obtained using meso-scale simulations and LES reveals alignment in the spatial patterns of erosion-related parameters, confirming that meso-scale simulations produce satisfactory atlases where regional differences are consistently captured with LES.

We have discussed the distinctions between LES and meso-scale simulations, contextualizing them within the framework of current literature as outlined below:

*By conducting 10-year long meso-scale simulations to account for long-term climatology, our study has highlighted the variation of rain-induced erosivity across the Dutch North Sea.*

*Examining the portion of the Dutch North Sea covered in the reanalysis-based erosion atlas by Hannesdóttir et al. (2024b), no clear trends can be inferred within this region. This is because their atlas focuses on Scandinavian regions and only marginally and partially covers the Dutch North Sea. According to this atlas, the incubation period is approximately 4 years over the covered Dutch North Sea, which is about 40% lower than the incubation period resulting from our meso-scale based erosion atlas. These differences can be attributed not only to the weather model with specific resolutions and periods but also to assumptions regarding DSD, drop falling speed, damage model, and fatigue characteristics of the LEP system. Such assumptions have a dramatic effect on the resulting incubation period. Due to the complexity of the calculations behind an erosion atlas, considering the assumptions and models used to generate it, it is not possible to detail where the differences between our atlas and the one by Hannesdóttir et al. (2024b) come from. Future research should aim at dedicated comparisons of erosion atlases, systematically breaking down the calculation chain and comparing results for each portion.*

*Our study indicates that a LES setup with finer spatial and temporal resolution enhances the ability of simulations to capture more extreme events. This is because the smaller temporal resolution allows the simulations to detect more short-term extreme events with high rain intensity. Such high-intensity events contain larger and more erosive droplets. With larger temporal resolutions, these events are averaged out. Our comparative analysis with actual measurements reveals that both meso-scale and LES models tend to underpredict the accumulated damage. One reason is that the Marshall-Palmer distribution assumed by both numerical models significantly underestimates the amount of large droplets compared to what is measured. Another reason is that more extreme events are recorded than those simulated, especially at the instrumented offshore location. Significant uncertainties still exist in detecting such events in both measurements and simulations. Detailed measurements of rain in offshore locations are new, and further research is required to improve these measurements and establish confidence bounds.*

In the abstract, the authors claim to have validated their model runs using in-situ wind and rain observations. This would normally mean that the estimates of the models can be trusted within a certain error margin. However, the use of the word "validation" is here very questionable, because the estimates of the models are as much as several 100% different from the observations (see table 2).

We agree that this term is not appropriate in this case. We used the term validation to indicate the exercise of comparing measurements to simulations. We have replaced the term validate (or validation) with the term compare (or comparison).

In the text, the authors claim that the models capture trends of the erosion related parameters effectively, but this is not generally correct (for example, regarding the accumulated rain amount, which show different relative behaviors for the three sites in observations and simulations).

We have removed this claim and reformulated the results.

In summary, the authors do not validate the models. In the introduction, the authors mention that the LES simulations are performed to verify the meso-scale simulations. However, the meso-scale simulations show a better performance concerning wind modeling and the precipitation fields of the models are not compared in any meaningful way to justify the impression that the authors communicate regarding the LES model's superiority. No side-by side rain and wind fields are shown to document the LES model's strengths or weaknesses.

This has been addressed by reformulating the results and adding the new section that focuses on a detailed comparison between meso-scale simulations and high-resolution LES.

Interestingly, the authors state that the LES based approach is better at capturing extreme events, without showing this important aspect in the paper! Instead, they only show results regarding rain-intensity in the high-resolution simulations (Figure 2), but here they omit to show the results from the meso-scale simulations. This omission is problematic, because it is important to show the comparison for the simulation that the atlas is based on. In general, neither approach is well evaluated, and it is hard to judge the pros and cons of their methods based on the presented results.

This has been addressed by the new section that focuses on a detailed comparison between meso-scale simulations and high-resolution LES.

For erosion studies on wind turbines, a key parameter is the incubation period (IP), which is the life-time of the leading-edge-protection system. I find the low numbers in the last column in Table 2 here (and elsewhere) highly problematic, mainly because they are not discussed. The measurement data indicates that the IP at the offshore LEG site is little more than two years. The overall result of this study indicates that it would be significantly shorter in the planned wind farms in the North, which, in turn, would indicate that these wind farms' O&M costs would be enormous. It is important to discuss what these low numbers mean in terms of cost-efficient operation of wind farms. Do they put a question mark on all the North Sea wind exploitation plans? Or is this low number simply a reflection of the methods behind the IP estimation are incorrect?

As added in the discussion section:

*As previously mentioned, the estimates of incubation periods provided by this study, whether derived from measurements or simulations, are based on various methods and several assumptions. Unfortunately, these estimates have not been validated in real-world conditions yet. However, we can say that these figures are roughly in line with the leading edge repair interventions of wind turbines in the Dutch North Sea.*

The results in the Figures 3-5 are not very easy to understand, partly based on the absolute numbers used (with very many decimals). The authors should normalize the numbers such that deviations can be seen in per cent. Also, the results from both models should be shown.

We agree that normalizing the figures would indeed make it easier to identify differences, but it would remove the physical meaning of the numbers, which we want to convey for further comparison with alternative or future studies. By retaining the absolute figures, we can preserve their physical meaning while still being able to spot differences. Therefore, we have chosen to opt for the latter option.

Another reason for recommending the rejection of this paper is that the authors fail to report a conflict of interest, although they are presenting results from a commercial tool in a core area of the company that produces this tool. One of the authors is hired by this company and the other seems to have strong links to it via project funding. Hence, there is an obvious conflict of interest, and by not acknowledging it, the presented research cannot be judged in a transparent way by readers. The results regarding the LES simulations are of high scientific interest, but we can only take these results seriously if we trust the presented work. To acknowledge their obvious conflict of interest is a first step towards creating such trust. Many authors mistake the

existence of conflict of interests with scientific misconduct. However, it is only misconduct if the situation is not acknowledged (in a European context, this is elaborated here H2020 INTEGRITY - Conflict of interest in research: what is it and why it matters?).

Thank you for bringing this to our attention. Indeed, we overlooked this matter. We now acknowledge the conflicts of interest as follows:

*G. van D. is employed at Whiffle, the developer of ASPIRE, the weather model utilized in this study. M. C. and G. van D. conducted this research as part of their collaboration within the PRecipitation atlas for Offshore Wind blade Erosion Support System (PROWESS) project, which was funded by The Netherlands Enterprise Agency (RVO), part of the Dutch Ministry of Economic Affairs.*

Minor comments:

Lines 150-153: The authors write *However, it is essential to note that DSD measurements obtained from current sensors, such as disdrometers, remain quite uncertain (Letson and Pryor, 2023; Caboni et al., 2024; Asta Hannesdottir et al., 2024a). This is due to the fact that these sensors and their algorithms are typically optimized to accurately detect total precipitation amounts rather than the DSD itself.* This is not correct. The disdrometers are made to detect droplets; not optimized for rain rates, which is a derived parameter from the instrument (see Johannsen et al 2020). Precipitation measurement from different types of disdrometers also vary (Angulo-Martinez et al 2018) .

Johannsen LL, Zambon N, Strauss P, Dostal T, Neumann M, Zumr D, Cochrane TA, Blöschl G, Klik A. Comparison of three types of laser optical disdrometers under natural rainfall conditions. Hydrol Sci J. 2020 Jan 21;65(4):524-535. doi: 10.1080/02626667.2019.1709641.

Angulo-Martínez, M., Beguería, S., Latorre, B., and Fernández-Raga, M.: Comparison of precipitation measurements by OTT Parsivel[2] and Thies LPM optical disdrometers, Hydrol. Earth Syst. Sci., 22, 2811–2837, https://doi.org/10.5194/hess-22-2811-2018, 2018.

We have elaborated on our statements in the introduction as follows:

*Disdrometers indeed detect droplets, and the rain amount is derived from the DSD. However, disdrometers cannot measure the full spectrum of droplet sizes and therefore rely on algorithms that make assumptions to improve the estimation of the rain amount. In our experience this seems to be rather site/precipitation dependent.*

Figure 7: The legends should not block the data shown.

The legend blocks only marginal data, which is not essential for the figure's purpose, which is to compare measurements with simulations.

---

## Author Comment (AC2)

**Response to Referee 2**

Developing an atlas of rain-induced leading edge erosion for wind turbine blades in the Dutch North Sea (wes-2024-174)

**Dear Reviewer,**

Thank you for reviewing our article. In response to Referee 1's comments, we have made significant revisions to the paper, addressing your specific suggestions and proposed changes too. Below, you will find details on the modifications related to your comments.

**Sincerely,**

**Marco Caboni and Gerwin van Dalum**

**Specific comments and proposed revisions:**

-The weather modeling used captures large-scale meteorological trends and spatial patterns in wind and precipitation, especially the northeast-southwest gradient in the Dutch North Sea. The paper clearly states that both LES and mesoscale simulations compared to rainfall experimental measurements underestimate accumulated LEE damage related to a referenced wind turbine, particularly due to underrepresentation of extreme events and large raindrops. The summary rightly identifies the Marshall–Palmer distribution's inability to represent large droplets (>3 mm), which is a key limitation discussed in the paper.

-In one hand, authors employed Whiffle's ASPIRE model, a GPU-accelerated large-eddy simulation (LES) and mesoscale weather modeling platform, to simulate weather conditions over two timeframes: a high-resolution one-year LES (2022–2023) for validation purposes and a lower-resolution ten-year mesoscale simulation (2014–2023) for long-term trend analysis. These simulations were compared against real experimental measurements of wind and rainfall collected at offshore, coastal, and onshore sites in the Netherlands. While the LES simulations more accurately captured extreme weather events and aligned better with observational data, the mesoscale simulations were deemed sufficient for long-term trend analysis and atlas development. However, an important result is that both simulation types underestimated accumulated damage compared to measurements. This discrepancy is primarily due to two factors: the underrepresentation of large raindrops in the Marshall-Palmer drop size distribution used in the model, and the lower frequency of extreme events in the simulations compared to observations.

---The authors should clearly identify the novelty of this result and explain how LES and mesoscale simulations serve as complementary tools in atmospheric research, each appropriate for different types of studies and objectives.

We have included a new section that focuses on a detailed comparison between meso-scale simulations and high-resolution LES performed over 1 year over the Dutch North Sea. In this section, we specifically compare annual rainfall and total annual damage, categorized into bins based on wind speed and rain rate, and analyze the contour maps. The year-long comparison of meso-scale simulations and high-resolution large eddy simulations (LES) at selected sites in The Netherlands showed that the accumulated damage estimated from meso-scale simulations is 7 to 20% lower than that obtained from LES. These differences can be attributed to the LES setup's ability to capture more extreme events due to its finer spatial and temporal resolution. Moreover, the side-by-side comparison of the contour maps obtained using meso-scale simulations and

LES reveals alignment in the spatial patterns of erosion-related parameters, confirming that meso-scale simulations produce satisfactory atlases where regional differences are consistently captured with LES.

We have discussed the distinctions between LES and meso-scale simulations, contextualizing them within the framework of current literature as outlined below:

By conducting 10-year long meso-scale simulations to account for long-term climatology, our study has highlighted the variation of rain-induced erosivity across the Dutch North Sea. Examining the portion of the Dutch North Sea covered in the reanalysis-based erosion atlas by Hannesdóttir et al. (2024b), no clear trends can be inferred within this region. This is because their atlas focuses on Scandinavian regions and only marginally and partially covers the Dutch North Sea. According to this atlas, the incubation period is approximately 4 years over the covered Dutch North Sea, which is about 40% lower than the incubation period resulting from our meso-scale based erosion atlas. These differences can be attributed not only to the weather model with specific resolutions and periods but also to assumptions regarding DSD, drop falling speed, damage model, and fatigue characteristics of the LEP system. Such assumptions have a dramatic effect on the resulting incubation period. Due to the complexity of the calculations behind an erosion atlas, considering the assumptions and models used to generate it, it is not possible to detail where the differences between our atlas and the one by Hannesdóttir et al. (2024b) come from. Future research should aim at dedicated comparisons of erosion atlases, systematically breaking down the calculation chain and comparing results for each portion.

Our study indicates that a LES setup with finer spatial and temporal resolution enhances the ability of simulations to capture more extreme events. This is because the smaller temporal resolution allows the simulations to detect more short-term extreme events with high rain intensity. Such high-intensity events contain larger and more erosive droplets. With larger temporal resolutions, these events are averaged out.

---However, despite effectively reproducing these trends, a significant limitation of the study that should be clarified is its systematic underestimation of absolute leading-edge erosion (LEE) damage.

Currently, we lack sufficient information to explain why both meso-scale and LES models tend to underpredict the accumulated damage compared to actual measurements. Besides uncertainties in the simulations, we cannot rule out that these differences may also result from significant uncertainties affecting the measurements (see Caboni et al., 2024). In the discussion section, we acknowledge the uncertainties present in both the measurements and simulations as follows:

Our comparative analysis with actual measurements reveals that both meso-scale and LES models tend to underpredict the accumulated damage. One reason is that the Marshall-Palmer distribution assumed by both numerical models significantly underestimates the amount of large droplets compared to what is measured. Another reason is that more extreme events are recorded than those simulated, especially at the instrumented offshore location. Significant uncertainties still exist in detecting such events in both measurements and simulations. Detailed measurements of rain in offshore locations are new, and further research is required to improve these measurements and establish confidence bounds.

-In the other hand, the erosion model used in the study estimates the incubation period—the time before visible erosion begins—based on ASTM regression equations and assumes a

polyurethane leading edge protection (LEP) system on a 15 MW reference wind turbine. The results reveal a clear spatial variation in erosion risk across the Dutch North Sea. The estimated incubation period ranges from 8–9 years in the southwest to 6–7 years in the northeast. This variation is attributed to higher average wind speeds and greater rainfall in the northeastern regions. The study concludes that although there are uncertainties in absolute damage estimation, the rainfall simulations effectively capture spatial trends in erosion risk due to weather conditions.

---The authors should clearly outline the aspects of their work that have been validated: whether it is the comparison of rainfall observations to simulations, or the progression of erosion damage, which is not currently depicted in the work.

In our study, we did not validate either the weather model or the erosion model. As part of our paper's scope, we compared weather simulations to measurements. Given the significant uncertainties and differences observed, we believe that the term "validation" is not appropriate in this context. Therefore, we have replaced "validate" (or "validation") with "compare" (or "comparison").

-Moreover, the use of normalized incubation resistance (NOR) derived from rotating-arm rain erosion tests (Slot et al., 2025) may not fully reflect the complex real-world conditions encountered offshore, potentially contributing to the observed discrepancies. The paper emphasizes that while the rainfall model captures trends, the absolute values of damage remain uncertain without direct validation from experimental observations in operational wind turbines. Moreover, authors state future work will focus on improving the representation of drop size distributions and fall velocities in the ASPIRE model and exploring real-time erosion forecasting to enable adaptive turbine operation during extreme weather events.

---The authors should clarify first how rainfall simulations relate to damage progression uncertainty analysis for future work, as this is not detailed or validated in the paper.

In the discussion section, we have addressed the uncertainties of our approach as follows:

Our comparative analysis with actual measurements reveals that both meso-scale and LES models tend to underpredict the accumulated damage. One reason is that the Marshall-Palmer distribution assumed by both numerical models significantly underestimates the amount of large droplets compared to what is measured. Another reason is that more extreme events are recorded than those simulated, especially at the instrumented offshore location. Significant uncertainties still exist in detecting such events in both measurements and simulations. Detailed measurements of rain in offshore locations are new, and further research is required to improve these measurements and establish confidence bounds.

As previously mentioned, the estimates of incubation periods provided by this study, whether derived from measurements or simulations, are based on various methods and several assumptions. Unfortunately, these estimates have not been validated in real-world conditions yet. However, we can say that these figures are roughly in line with the leading edge repair interventions of wind turbines in the Dutch North Sea.

-Additionally, the simplified linear damage accumulation approach adopted by the incubation period (IP) model (Caboni et al., 2024), based on the Palmgren–Miner rule, may not accurately capture non-linear effects associated with severe meteorological events. As a result, without direct validation through observed erosion damage on operational wind turbine blades, these

predictions related to material damage evolution remain theoretical. The reliability and practical applicability of these estimations are consequently limited, underscoring the critical need for direct comparison with actual erosion observations from operational offshore wind farms under comparable environmental conditions to validate and refine the predictive capabilities of the proposed model.

---The authors should consider these limitations and clearly state the scope of the proposed study in the paper.

The limitations of the approach are discussed in the discussion section as mentioned above. Related to the approach limitations, future work is outlined in the conclusion section as follows:

Future work will involve implementing a more representative drop size distribution and fall velocity in the weather model. In this context, models will need to be validated with more reliable measurements, which is also a topic of ongoing and future research.